# Tailoring of the Structural, Optical, and Electrical Characteristics of Sol-Gel-Derived Magnesium-Zinc-Oxide Wide-Bandgap Semiconductor Thin Films via Gallium Doping

**DOI:** 10.3390/ma16196389

**Published:** 2023-09-25

**Authors:** Chien-Yie Tsay, Shih-Ting Chen, Hsuan-Meng Tsai

**Affiliations:** Department of Materials Science and Engineering, Feng Chia University, Taichung 40724, Taiwan; sarah15937@gmail.com (S.-T.C.); a23585050@gmail.com (H.-M.T.)

**Keywords:** wide-bandgap oxide semiconductor, MgZnO thin film, sol-gel spin coating, photoluminescence emission, electrical properties

## Abstract

The Ga-doped Mg_0.2_Zn_0.8_O (GMZO) transparent semiconductor thin films were prepared using the sol-gel and spin-coating deposition technique. Changes in the microstructural features, optical parameters, and electrical characteristics of sol-gel-synthesized Mg_0.2_Zn_0.8_O (MZO) thin films affected by the amount of Ga dopants (0–5 at%) were studied. The results of grazing incidence X-ray diffraction (GIXRD) examination showed that all as-prepared MZO-based thin films had a wurtzite-type structure and hexagonal phase, and the incorporation of Ga ions into the MZO nanocrystals refined the microstructure and reduced the average crystallite size and flatness of surface roughness. Each glass/oxide thin film sample exhibited a higher average transmittance than 91.5% and a lower average reflectance than 9.1% in the visible range spectrum. Experimental results revealed that the optical bandgap energy of the GMZO thin films was slightly higher than that of the MZO thin film; the Urbach energy became wider with increasing Ga doping level. It was found that the 2 at% and 3 at% Ga-doped MZO thin films had better electrical properties than the undoped and 5 at% Ga-doped MZO thin films.

## 1. Introduction

Wide-bandgap elemental semiconductor diamond (Eg = 5.5 eV) and compound semiconductors, β-Ga_2_O_3_ (Eg = 4.9 eV), GaN (Eg = 3.4 eV), and ZnO (Eg = 3.35 eV), combine visible light transparency and high carrier mobility as well as high-temperature operational stability and a strong power-handling capability, and are usually applied in ultraviolet (UV) photodetectors, high-power electron devices, and transparent electronics [1,2,3,4]. The wurtzite structure Mg_x_Zn_1−x_O (x ≤ 0.25) is a wide-bandgap ternary oxide semiconductor system with unique characteristics, such as intrinsic blindness to the visible light and radiation toughness, as well as low-temperature synthesis and wet-etch availability [5,6]. It is an optimal candidate for use in optoelectronic devices, including ultraviolet thin-film phototransistors and photodetectors [7,8]. Several studies have shown that Mg_0.2_Zn_0.8_O thin films exhibit device-quality physical properties and have been used as active layers in photoelectric devices. Our previous study revealed that the Mg_0.2_Zn_0.8_O thin film had the best transparency of 92% and the lowest surface root mean square (RMS) of 1.63 nm among the sol-gel-deposited Mg_x_Zn_1−x_O (x = 0–0.36) thin films. It was successfully utilized as the active channel layer of bottom-gate structured thin-film transistors (TFTs), which exhibited n-channel behavior and operated in the enhancement mode, and had an on-to-off current ratio greater than 10^7^ [9]. Yu et al. reported that the measured photoelectric characteristics showed that sol-gel-synthesized MgZnO nanoparticles (molar ratio of Mg to Zn was adjusted to 1:4 for MgZnO sols) were suitable for use as the absorption layer of metal–semiconductor–metal (MSM)-structured ultraviolet (UV) photodetectors [10]. Mg_0.2_Zn_0.8_O nanocrystalline thin films were synthesized using the sol-gel method, and these oxide thin films exhibited wide UV properties, covering the solar blind to the near-UV range, as reported by H. Liu et al. [11]. K.W. Liu and colleagues prepared Mg_0.2_Zn_0.8_O thin films by radio frequency (RF) magnetron sputtering the fabricated Mg_0.2_Zn_0.8_O MSM UV photodetectors with a low dark current (7 nA at 5 V bias) and fast photoresponse time (10 ns) [12]. 

The modification of the chemical composition of oxide semiconductor thin films, such as adjusting the cation ratio or incorporating foreign ions, is a simple, effective, and widely used method for improving visible-light transmission, tailoring the optical bandgap, and enhancing the electrical properties. The trivalent element gallium (Ga) is one of the most promising candidates as an n-type dopant, mainly because of the use of Ga substitution for Zn (Ga_Zn_, donor defect) to enhance the electrical conductivity of ZnO-based films for the application of high-performance optoelectronic devices [13,14]. Kim et al. proposed that the incorporation of Ga into the hexagonal wurtzite lattice of ZnO crystals could attenuate interfacial charge transfer between Ga-doped ZnO nanocrystals because the Fermi level of ZnO nanocrystals is increased by increasing Ga dopants [15]. Xie et al. demonstrated that Ga doping improves n-type conductivity by two orders of magnitude through electrical, optical, and structural studies in cubic MgZnO films grown on sapphire substrates via metal-organic chemical vapor deposition (MOCVD) [16]. In addition, Ga-doped MgZnO thin films have been utilized as transparent conducting oxide layers in thin-film photovoltaics [17,18,19]. 

Various deposition techniques have been utilized to prepare the transparent MgZnO thin films, such as metal-organic chemical vapor deposition (MOCVD), molecular beam epitaxy (MBE), radio-frequency (RF) magnetron sputtering, pulsed laser deposition (PLD), the spray pyrolysis technique, the sol-gel method, and so on [10,20]. The sol-gel method is an easy-to-implement approach for large-scale, functional oxide thin-film deposition because it does not require costly equipment and components for a high-vacuum system. In addition, it provides the efficient incorporation of dopants’ impurity to flexibly adjust the chemical composition of oxide thin films to modulate physical properties [21,22]. In this study, we investigated the effects of Ga doping on the microstructural features, optical characteristics and electrical properties, and photoluminescence emission characteristics of Mg_0.2_Zn_0.8_O semiconductor thin films prepared using sol-gel and spin-coating techniques. 

## 2. Materials and Methods

Undoped and Ga-doped Mg_0.2_Zn_0.8_O (MZO and GMZO) films were grown on alkali-free glasses (Nippon Electric Glass Co., Ltd., NEG OA-10G) using a sol-gel spin coating process. Analytical reagent-grade metal salts, including zinc acetate dihydrate [Zn(CH_3_COO)_2_·2H_2_O, J.T. Baker, Center Valley, PA, USA, 99.98%], magnesium nitrate hexahydrate [(Mg(NO_3_)_2_·6H_2_O, Fluka, Everett, WA, USA, 99.0%], and gallium nitrate hydrate [(Ga(NO_3_)_3_·*x*H_2_O, Aderich, St. Louis, MO, USA, 99.98%], were chosen as the starting materials for Zn, Mg, and Ga ions, respectively. The precursor solutions for the sol-gel oxide film coating were synthesized by dissolving metal salts in 2-methoxyethanol (2-ME, TEDIA, Fairfield, OH, USA, 99.9%) solvent and then adding a diethanolamine (DEA, Alfa Aesar, Haverhill, MA, USA, 99.0%) as a stabilizer of the mixed solution according to the stoichiometric ratio of the Ga-doped Mg_0.2_Zn_0.8_O sol. The molar ratio of metal ions to DEA was fixed at 1.0, and the concentration of the metal ions in each resulting solution was 0.5 M. The Ga doping concentration ([Ga]/[Mg] + [Zn]) in the resultant solution was varied from 0 to 5 at%. Transparent and clear sols were obtained by heating and stirring at 60 °C for 120 min using a magnetic stirrer on a ceramic hotplate. These as-synthesized solutions were then stored and aged at room temperature for 5 d prior to use as precursors for the spin-coating process. MZO-based sol films were deposited on glass substrates, which had been pre-cleaned, at 500 rpm for 10 s and then at 1500 rpm for 30 s. After deposition, each as-coated sol-gel film was first dried at 300 °C for 10 min to evaporate the solvents, decompose most of the organic polymers to form a sol film, and then annealed at 500 °C in a tube furnace under an air atmosphere for another 60 min to burn out residual organics, improve densification, and form a crystalline oxide film. 

The crystal structure and phase identification of the obtained Mg_0.2_Zn_0.8_O-based thin films were determined using a Bruker D8 Discover Multipurpose X-ray diffraction diffractometer (Billerica, MA, USA) with Cu-Kα radiation (λ = 0.1541 nm) using the glancing incidence technique at an incident angle of 0.8°. The microstructural features of the thin film samples were observed using a Hitachi S-4800 field-emission scanning electron microscope (FE-SEM; Hitachi High-Technology, Tokyo, Japan) and evaluated using cross-sectional micrographs. The free surface morphology and film surface roughness were characterized using a Digital Instruments NS4/D3100CL/MultiMode scanning probe microscope (SPM, Mannheim, Germany) in the tapping mode. The transmission spectra and reflectance spectra of the glass/MZO-based thin film samples were characterized and recorded using a Hitachi U-2900 ultraviolet-visible (UV-Vis) double-beam spectrophotometer (Hitachi High-Technology, Tokyo, Japan) in the wavelength range of 190–810 nm. Photoluminescence (PL) emission spectra were measured using a Horiba Jobin Yvon LabRAM HR Micro-PL spectrometer (Paris, France) equipped with a He-Cd laser at a wavelength of 325 nm and a power of 40 mW. The electrical properties, including the major carrier concentration density, carrier Hall mobility, and electrical resistivity, were measured with a Hall measurement system (Ecopia HMS-3000, Gyeonggi-do, Republic of Korea) using the van der Pauw method with a 4-point probe under an applied magnetic field of 0.55 Tesla (T). All physical property measurements of the oxide thin films were performed at room temperature. The detailed abbreviations used in the paper are listed in Table 1. 

## 3. Results and Discussion

The crystallinity and phase identification of the sol-gel-synthesized Mg_0.2_Zn_0.8_O (MZO) and Ga-doped Mg_0.2_Zn_0.8_O (GMZO) thin films were analyzed using X-ray diffraction using the glancing incidence technique in the scanning range of 2θ = 25° to 60°. Figure 1 shows the grazing incidence X-ray diffraction (GIXRD) patterns of MZO-based thin films with different Ga doping contents annealed at 500 °C in air. There are three major characteristic diffraction peaks in the low angle side corresponding to the (100), (002), and (101) planes and three weak diffraction peaks in the high angle side corresponding to the (102), (110), and (103) planes, of JCPDS card No. 36-1451 of zincite [23]. The intensities of the four different diffraction peaks for the MZO-based thin films exhibited the following order: I_(101)_ > I_(100)_ > I_(002)_ > I_(110)_. No preferential orientation was observed in these oxide films, which was attributed to the choice of the amorphous glass substrate and the nature of the solution-process deposition technique. In addition, there were no X-ray diffraction peaks or signals that corresponded to any other oxide or compound due to phase separation. These results indicate that all the as-synthesized MZO-based thin films are polycrystalline in nature and have a single-phase hexagonal wurtzite structure. 

The ionic radius of Ga^3+^ (0.62 Å) is smaller than those of Zn^2+^ (0.74 Å) and Mg^2+^ (0.72 Å), which suggests that it is easily incorporated into the crystal lattices of the MgZnO hexagonal wurtzite structure. In addition, the covalent band length of Ga-O (1.92 Å) is slightly shorter than those of Zn-O (1.97 Å) and Mg-O (2.06 Å) [15,17]. This was expected without significant lattice strain and small inter-planar spacing changes. It was found that three major GIXRD peaks were slightly broadened with increasing Ga doping concentration, and the bottom of the diffraction peaks of the (002) and (101) planes overlapped for Ga doping levels higher than 3 at% (spectra (iii) and (iv) of Figure 1). The GIXRD peak broadening results from the nanoscale effect and suggests that the crystallite sizes of the GMZO thin films were smaller than those of the MZO thin films. The overlap of the peaks can be attributed to the degradation of the crystallinity of the MZO thin films after their incorporation into impurity dopants. 

According to the results of Gaussian fitting for the three major diffraction peaks, the values of (FWHM) of the corresponding diffraction peaks increased with the increasing Ga dopant content. The crystallite sizes of the polycrystalline oxide thin films were estimated from the full width at half maximum (FWHM) and Bragg angle of the specific XRD diffraction peaks (including the (100), (002), and (101) peaks) and the X-ray wavelength using Scherrer’s formula [21]. The results of the estimates are summarized in Table 2. It can be seen that the average crystallite size decreased from 13.8 nm to 10.3 nm as the Ga dopant content increased from 0 to 5 at%. It can be inferred that increasing the Ga dopant content in the MZO nanocrystals led to a decrease in the average crystalline size owing to the lattice distortion effect and the decrease in the grain growth rate caused by impurity incorporation. The B-doped AZO sol-gel films also showed a similar tendency, as determined by the X-ray diffraction analysis [22]. After doping with Ga, the peaks of the (100) and (002) planes showed a slight shift feature, confirming their incorporation. The a-axis and c-axis lattice parameters were calculated using a = λ/$\sqrt{3}$ in θ and c = λ/sinθ, respectively. The calculated results showed that Ga doping in the MgZnO nanocrystals could increase the a-axis length from 3.254 to 3.267 Å and the c-axis length from 5.193 to 5.233 Å. This can be attributed to the Ga incorporation causing lattice strain. We found that the determined a-axis and c-axis lattice parameters for the MgZnO thin film were close to those of the polycrystalline ZnO, which were obtained from the standard JCPDS data (a = 3.249 Å and c = 5.206 Å). 

The cross-sectional FE-SEM images (Figure 2) of the as-grown MZO and various GMZO polycrystalline thin films showed identical morphologies with granular microstructures consisting of nano-sized particles and uniform film thickness. It can be seen from the micrographs that the size of the nanoparticles decreased with increasing external Ga doping content (the same results as those of the GIXRD study), and that small numbers of nanopores were presented in these sol-gel-synthesized MZO-based thin films. The microstructural defects associated with the nanoscale pores were attributed to the thermal decomposition of the coating precursors and residual organic matters in the sol-gel-derived oxide thin films. The thicknesses of the MZO-based thin films were estimated through the cross-sectional observation of the FE-SEM images. The mean thicknesses of the MZO and GMZO thin films were 70 and 73 nm, respectively. This difference can be attributed to changes in the viscosity of the coating solution after the addition of Ga metal salt. 

The film surface topography and root-mean-square (RMS) roughness were studied from the three-dimensional (3D) SPM micrographs taken from the free surfaces of each glass/thin film sample, as shown in Figure 3. They showed an observable granular configuration without micro-cracks or significant porosity. The dense surface microstructures consisted of tightly packed nano-sized particles that were regularly and uniformly distributed. Table 2 shows the measured surface RMS roughness values of the four MZO-based thin film samples. The SPM image of surface topography shows a relatively larger particle size, and a regular reduction in the particle size is observed with increasing Ga dopant content. The results of the Ga doping effect obtained from the cross-sectional view observation (FE-SEM images) and free surface morphology examination (SPM images) are in agreement with the crystalline value calculated from the GIXRD data using Scherrer’s formula. 

Figure 4 shows the optical transmittance spectra and reflectance spectra of the glass/oxide film samples measured at room temperature using a conventional ultraviolet-visible spectrometer to study the optical properties. Each oxide thin film sample exhibited a similar transmittance spectrum, which showed that the average transmittance was greater than 91.5% and the average reflectance was less than 9.1% in the wavelength range between 400 and 800 nm, as shown in the fifth and sixth rows of Table 2. In addition, there was a steep absorption edge at approximately 350 nm, and the transparency was close to zero at a light wavelength of less than 250 nm. The recorded reflectance of the GMZO thin films was lower than that of the MZO thin films in the UV wavelength band. This decrease is related to the fact that the front had a relatively flat free surface because it had finer microstructures and therefore exhibited less light scattering.

To determine the important optical parameters of the optical bandgap energy and Urbach energy, we calculated the absorption coefficient [α(λ)] of the four MZO-based thin films using the recorded optical transmittance [T(λ)] and optical reflectance [R(λ)] data according to the following equation [24]:α (λ) = 1/t ln [(1 − R^2^)/T],(1)
where t is the measured thickness of the oxide film specimen. The optical bandgap energy of semiconductors can be obtained using Tauc’s relation ((αhν) 1/n = C × (hν − Eg), where hν is the photon energy of incident light, C is the proportionality constant, Eg is the optical bandgap energy, and n = ½ and 2 for direct and indirect bandgap semiconductors, respectively) and the Tauc plot method [25]. The curves of the square of the absorption energy (αhν)^2^ versus the photon energy (hν), also known as the Tauc plot, are shown in Figure 5a. According to Tauc’s relation, if (αhν)^2^ = 0, Eg = hν. The optical bandgap energy was obtained by extrapolating the linear region of the Tauc plot to the horizontal (photon energy) axis [25]. The determined optical bandgap (Eg) of the MZO thin film was 3.58 eV, while the Eg of the GMZO thin films was slightly higher (0.1~0.2 eV) than that of the MZO thin film (Table 2). The same optical bandgap energy (Eg = 3.58 eV) for sol-gel-derived wurtzite Mg_0.2_Zn_0.8_O thin films was also reported by Ogawa and Fujihara [26]. In our previous study, the Eg of sol-gel-synthesized ZnO thin film was determined to be 3.25 eV [8]. The developed MZO thin films have bandgap shifts up to 0.33 eV compared to the pristine ZnO thin films.

The Urbach energy (E_u_) is used to study the effects of structural disorder and defect-state concentration in the thin films of oxide semiconductors [27]. The relationship between the absorption coefficient (α(λ)), photo energy (hν), and Urbach energy (E_u_) can be determined using the following equation [28]: α (λ) = α_0_ exp (hν/E_u_), (2)
where α_0_ is a constant. A plot of the dependence of the natural logarithm of the absorption coefficient (ln (α)) on the incident photon energy (hν) near the band edge is shown in Figure 5b. The Urbach energy was obtained by calculating reciprocal of the slope of the linear part of the absorption curve. The determined Urbach energy monotonically increased from 149 to 204 meV with increasing Ga dopant content (Table 2). This is attributed to the degenerating film crystallinity. The Eu of 1 at% Ga doped M_0.2_Z_0.8_O thin film is slightly lower than that of the 2 at% Al-doped ZnO thin film (165.6 meV) [22]. 

To gain further insight into the effect of Ga doping on the crystallinity and lattice defect state of the MZO thin films, the room-temperature photoluminescence (PL) emission spectrum of each MZO-based thin film was measured. Both sharp and weak and broad PL emission signals were detected in the as-prepared oxide thin films, as shown in Figure 6. The sharp emission peak in the UVA range (347.3 nm) corresponds to the near-band-edge emission (NBE), which was ascribed to radiation excited recombination, whereas the broad emission band in the green light range (540–580 nm) was related to the presence of deep level defects due to impurity dopants and/or intrinsic point defects in these oxide thin films [29,30]. We found that the peak wavelength of NBE emission was maintained at the same position, while the emission intensity decreased with increasing Ga dopant content, which can be attributed to the degraded crystal quality and increased defect density [31]. 

The electrical properties, including the electron concentration (n), Hall mobility (μ), and resistivity (ρ), of the four MZO-based thin films were measured using Hall effect measurement and are shown in Figure 7. The measured results confirmed that all oxide thin films exhibited n-type conductivity. The mean electron concentration (n) increased from 4.40 × 10^13^ cm^−3^ to 2.28 × 10^14^ cm^−3^ with increasing Ga doping content from 0 to 3 at% and then decreased to 5.46 × 10^13^ cm^−3^ when the Ga doping content reached to 5 at%. The electron concentration of the 3 at% Ga-doped MZO thin film is close to the reported electron concentration of the pristine ZnO sol-gel film (3.49 × 10^14^ cm^−3^) [22]. The 3 at% Ga-doped sample had the lowest resistivity of 1.28 × 10^3^ Ω cm, and the magnitude of resistivity for the 5 at% Ga doped sample is close to that of the undoped sample. This is because the electrical resistivity of semiconductors is approximately inversely proportional to the main carrier concentration density. Moreover, the 3 at% Ga-doped sample had the highest Hall mobility of 24.7 cm^2^/Vs in the present study. It is established that the Hall mobility is a combination of the mobility of impurity scattering and the mobility of grain boundary scattering. Therefore, among all the MZO-based samples, the oxide sample with the finest microstructure exhibited the lowest Hall mobility. According to the above discussion of the physical characteristics of the oxide thin films, Ga doping levels between 1 and 3 at% for M_0.2_Z_0.8_O transparent semiconductor thin films are suitable for use as the active channel layer of thin-film UV phototransistors.

## 4. Conclusions

Device-quality Ga-doped Mg_0.2_Zn_0.8_O (GMZO) transparent semiconductor thin films have been synthesized via the sol-gel spin-coating process on alkali-free glass substrates. The developed polycrystalline MZO and GMZO thin films had a monophase hexagonal wurtzite structure, and structural analyses revealed that the corresponding oxide thin films were composed of nanocrystalline grains. The optical bandgaps estimated from the recorded light transmittance and reflectance spectra for the GMZO thin films showed a slight blue shift compared to that of the MZO thin film. It has been shown that light Ga doping (< 5 at%) can improve the n-type conductivity of wurtzite MZO semiconductor thin films. In the present study, MZO thin films doped with 1–3 at% Ga dopants could achieve an electron concentration higher than 1.0 × 10^14^ cm^−3^, a Hall mobility faster than 21.0 cm^2^/Vs, and a resistivity lower than 1.0 × 10^3^ Ω cm. They have great potential for use in UV phototransistors with good performance.

## Figures and Tables

**Figure 1 materials-16-06389-f001:**
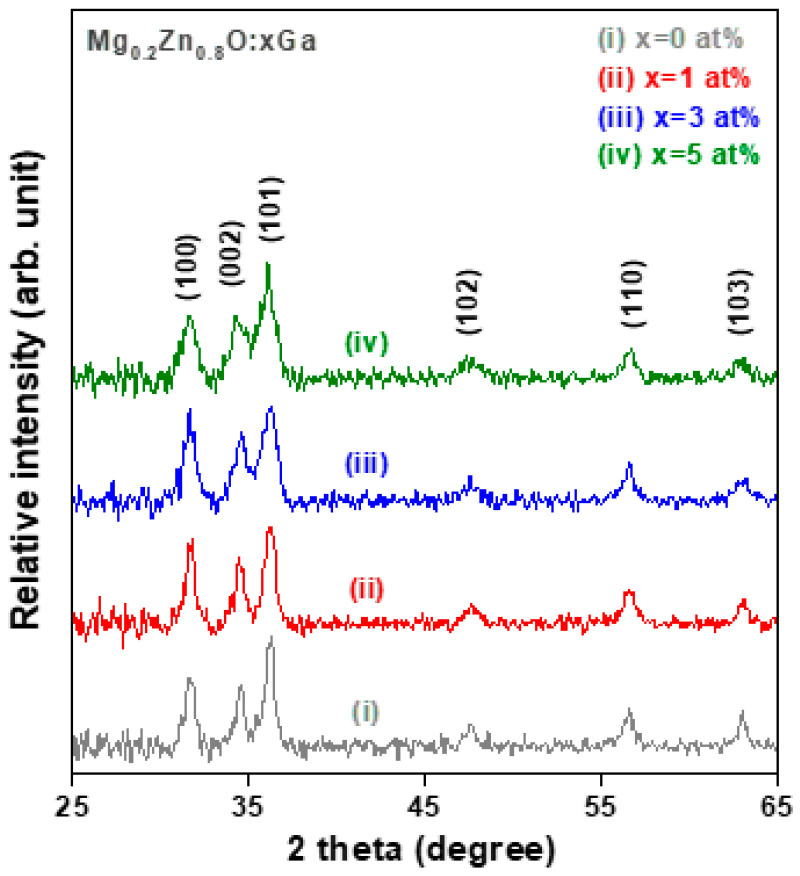
Grazing incidence X-ray diffraction (GIXRD) patterns of sol-gel-synthesized Mg_0.2_ZnO_0.8_ (MZO) thin films with different Ga doping concentrations.

**Figure 2 materials-16-06389-f002:**
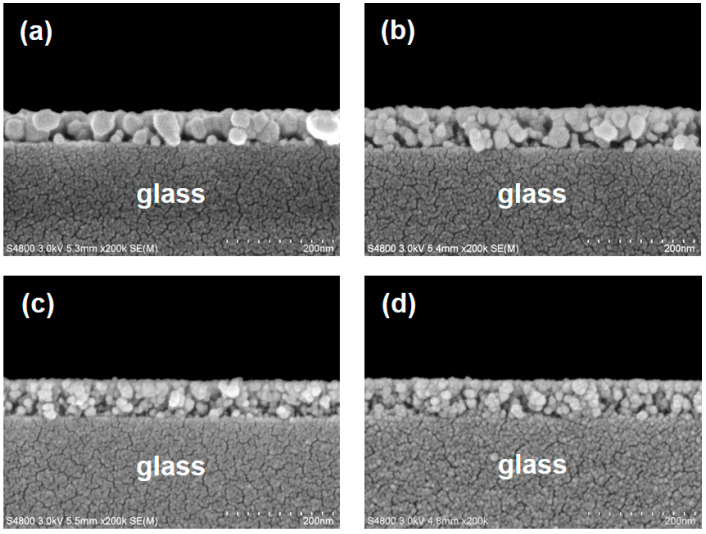
Cross-sectional scanning electron microscope (SEM) images of Ga-doped Mg_0.2_ZnO_0.8_ (GMZO) thin films deposited on NEG OA-10 glass substrates: (**a**) undoped, (**b**) 1 at%, (**c**) 3 at%, and (**d**) 5 at% Ga-doped thin film samples.

**Figure 3 materials-16-06389-f003:**
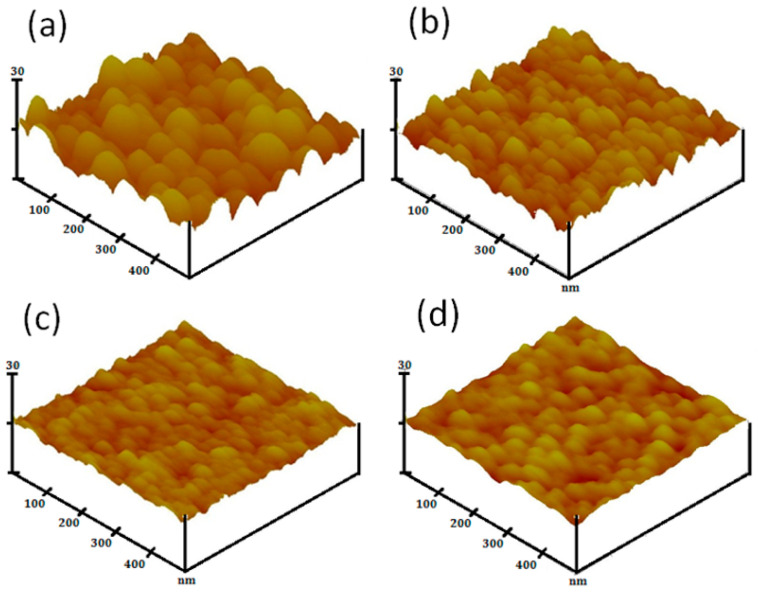
Scanning probe microscope (SPM) images of the free surface of GMZO thin film samples: (**a**) undoped, (**b**) 1 at%, (**c**) 3 at%, and (**d**) 5 at% Ga-doped thin films.

**Figure 4 materials-16-06389-f004:**
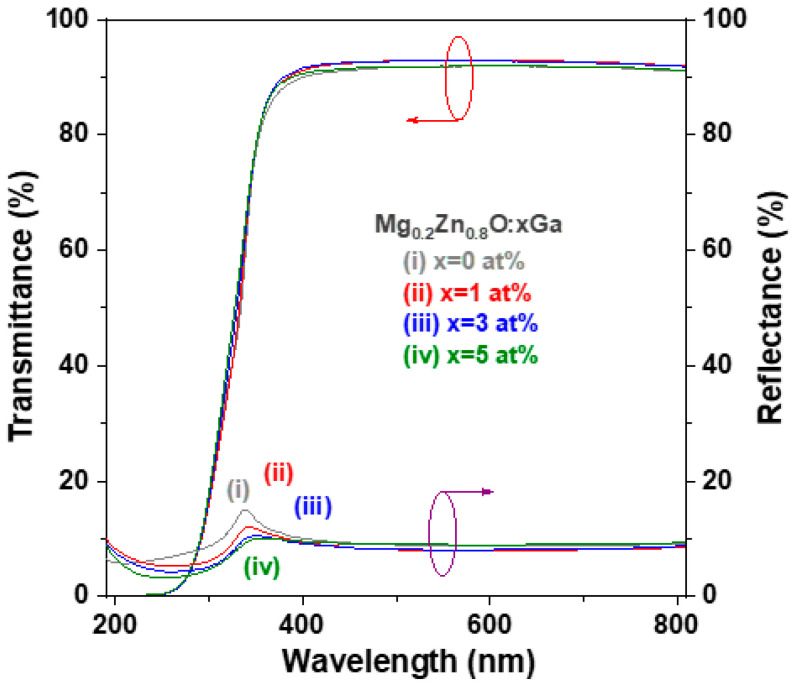
Optical transmission and reflection spectra of four prepared GMZO thin film samples.

**Figure 5 materials-16-06389-f005:**
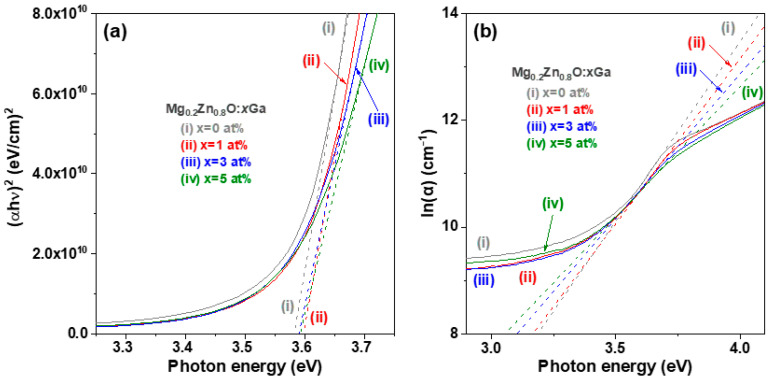
(**a**) Plot of (αhν)^2^ versus photon energy (hν) and (**b**) plot of ln(α) versus photon energy (hν) for the corresponding GMZO thin film samples.

**Figure 6 materials-16-06389-f006:**
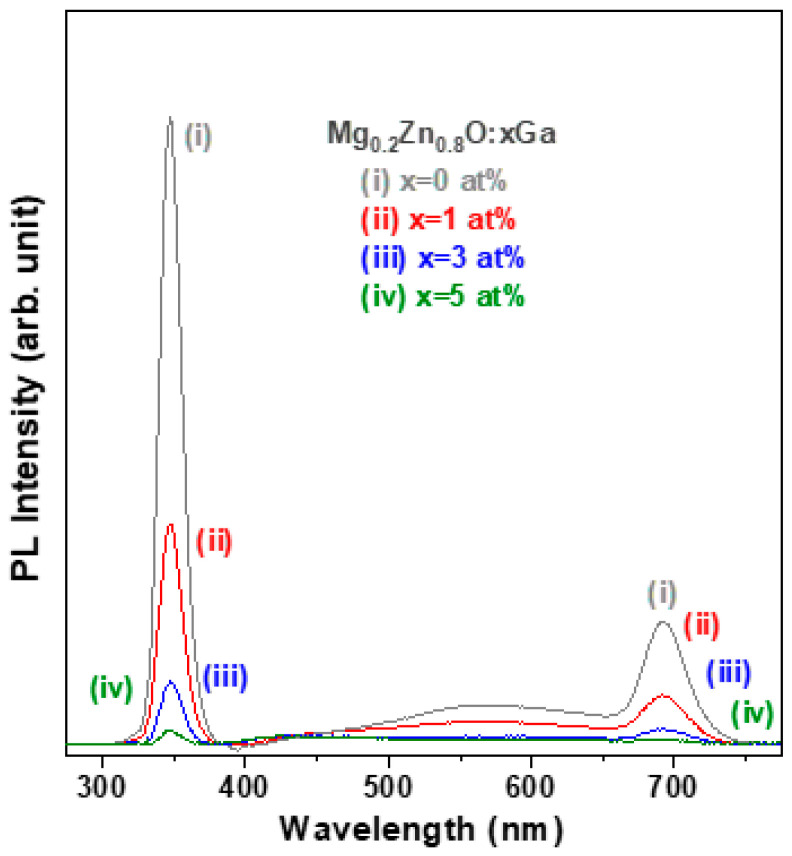
Room-temperature photoluminescence (PL) spectra of four GMZO thin films.

**Figure 7 materials-16-06389-f007:**
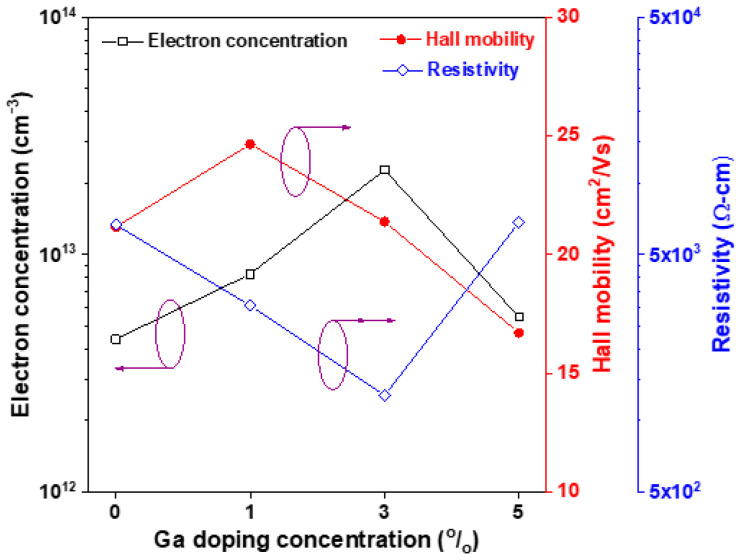
Variation in mean electron concentration (n), Hall mobility (μ), and resistivity (ρ) of GMZO thin films with Ga doping concentration.

**Table 1 materials-16-06389-t001:** List of abbreviations used in this paper (in alphabetical order).

Acronym	Stands	Acronym	Stands
2-ME	2-methoxyethanol	3D	three-dimensional
DEA	Diethanolamine	FE-SEM	field-emission scanning electron microscope
FWHM	full width at half maximum	GIXRD	grazing incidence X-ray diffraction
GMZO	Ga-doped Mg_0.2_Zn_0.8_O	MBE	molecular beam epitaxy
MOCVD	metal-organic chemical vapor deposition	MSM	metal–semiconductor–metal
MZO	Mg_0.2_Zn_0.8_O	NBE	near-band-edge emission
PL	Photoluminescence	PLD	pulsed laser deposition
RF	radio frequency	RMS	surface root mean square
SPM	scanning probe microscope	TFTs	thin-film transistors
UV	ultraviolet	UV-Vis	ultraviolet-visible

**Table 2 materials-16-06389-t002:** Comparison of structural characteristics and optical properties of sol-gel-grown Ga-doped Mg_0.2_Zn_0.8_O (GMZO) thin films.

Ga Doping Level (at%)	0	1	3	5
Average crystallite size (nm)	13.8	12.3	10.6	10.3
a-axis lattice parameter (Å)	3.254	3.255	3.255	3.267
c-axis lattice parameter (Å)	5.192	5.201	5.197	5.233
Root mean square roughness (nm)	3.9	2.3	1.2	1.0
Average optical transmittance (%) ^a^	91.57	92.59	92.54	91.67
Average optical reflectance (%) ^b^	8.98	8.26	8.34	9.06
Optical bandgap (eV)	3.58	3.60	3.59	3.59
Urbach energy (meV)	149	162	184	204

The average transmittance ^a^ and average reflectance ^b^ were calculated from the recorded transmittance and reflectance data of wavelengths from 400 to 800 nm.

## Data Availability

Data sharing is not applicable to this article.

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
