# Peer review of "Tailoring of the Structural, Optical, and Electrical Characteristics of Sol-Gel-Derived Magnesium-Zinc-Oxide Wide-Bandgap Semiconductor Thin Films via Gallium Doping"

_materials, 2023, doi:10.3390/ma16196389_

Round 1

Reviewer 1 Report

In this work, Tsay et al. introduce the results of a work evaluating the main properties of Ga-doped Zinc-Magnesium-Oxide, produced by sol-gel technique, for the fabrication of optoelectronic devices operating in the UV wavelength range.
The overall quality of the manuscript is good, methodologies are accurately described and results are properly discussed.
However, there are some major issues to be addressed to improve the scientific soundness of the paper, in my opinion, mostly aimed at justifying the possible use of GZMO for UV devices:

1. When  introducing a new semiconductor for UV devices with “unique characteristics” (as stated by the authors), a short paragraph should be added in the introductory section on a comparison with the most significant wide-bandgap semiconductors reported in the literature, such as diamond (see Carbon 189, 27-36, 2022), gallium oxide (see Journal of Materials Chemistry C 7, 8753-8770, 2019), zinc oxide (see ACS Appl. Mater. Interfaces 11, 26127–26133, 2019). I suggest the authors to elaborate a little bit more on this point, highlighting the “uniqueness” of GMZO with respect to the above mentioned materials for UV devices.
2. The novelty of the work is not completely clear. I mean, the real novelty is the use of sol-gel method for the production of GMZO films, and not the Ga-doping of MZO films itself, which has been already reported in the literature. As a consequence, the title of the paper could be misleading. I suggest to clearly mention the sol-gel deposition technique in the title, and to better highlight the novelty of the work in the introductory section.
3. The introduced material is specifically thought for UV photodetectors and phototransistors. However, no characterisation of the charge transport properties (e.g. photocurrent production under UV radiation, which is essential to validate a material for UV devices) has been reported in the manuscript. Purely optical (or electrical) characterisation is not enough to qualify a material for optoelectronics. Authors are strongly recommended to add some results, if available, on the responsivity of GMZO under UV radiation (or on its quantum efficiency) at different Ga concentrations, aimed at demonstrating that the material is not only able to efficiently absorb UV energy, but also to turn it into exploitable photocurrent.

English is ok.
Just a few typos to be corrected.

Reviewer 2 Report

In this manuscript, the authors studied the effect of Ga doping on the structural, optical, and electrical characteristics of zinc-magnesium-oxide wide bandgap semiconductor thin films. The manuscript should be accepted after addressing the following issues;

1.      In the material and method section, the authors should add the details of the source of chemicals.

2.      In XRD data, the authors should add more information related to the XRD, such as crystalline size, and lattice parameters.

3.      The authors should add Rietveld refinement of X-ray diffraction.

4.      The authors measured the electrical properties and secondly claimed that the film was without micro-cracks or significant porosity. Did the authors check the leakage current?

5.      The authors only mentioned the electron density; what about the hole mobility?

6.      There are too many abbreviations are used in this article. I recommend making a table to introduce all the abbreviations so that readers understand easily.

7.      What you did in this article should be written in the abstract. Reader is confuse what you actually did or what is your part in this part.

8.      Sometimes, the authors wrote Figure, but not every time. You should follow a single pattern throughout the paper.

9.      Table 1 is very informal; I recommend revisiting the table, as it is very important for this research work. Add previous values from the literature to compare the results.

10.  Many spelling and formatting typos in this paper, and the authors should check and revise them thoroughly        

Moderate editing of English language required

Round 2

Reviewer 1 Report

I’m happy with the revised version of the manuscript.

Authors have welcomed my suggestions and answered satisfactorily to my comments, improving the scientific soundness of the paper.

Reviewer 2 Report

Accepted in the present form. 

Okay